# A New Conditionally Immortalized Nile Tilapia (*Oreochromis niloticus*) Heart Cell Line: Establishment and Functional Characterization as a Promising Tool for Tilapia Myocarditis Studies

Yanghui Chen [1], Yuan Li [2], Dongneng Jiang [1], Defeng Zhang [3], Yu Huang [1], Jia Cai [1], Jichang Jian [1] and Bei Wang [1,*]

1 Guangdong Provincial Key Laboratory of Aquatic Animal Disease Control and Healthy Culture & Key Laboratory of Control for Diseases of Aquatic Economic Animals of Guangdong Higher Education Institutes, Fisheries College of Guangdong Ocean University, Zhanjiang 524088, China
2 Henry Fok College of Biology and Agriculture, Shaoguan University, Shaoguan 512005, China
3 Laboratory of Fishery Drug Development, Pearl River Fisheries Research Institute, Chinese Academy of Fishery Sciences, Ministry of Agriculture and Rural Affairs, Guangzhou 510380, China
* Correspondence: wong19820204@126.com or wangb@gdou.edu.cn; Tel.: +86-075-9238-3509

**Abstract:** A new cell line named the tilapia heart cell line (TAH-11) was established from the heart of tilapia (*Oreochromis niloticus*) by enzymatic digestion and mechanical separation. The TAH-11 cell line has been stably subcultured for over 80 generations and resuscitated after being frozen in liquid nitrogen for six months, with exuberant cell growth. The optimal culture condition of TAH-11 is L-15 medium supplemented with 10% fetal bovine serum at 25 °C. Sequencing of the mitochondrial 18S rRNAs confirmed that TAH-11 cells were derived from the tilapia. TAH-11 was also identified as a myocardial cell line based on the mRNA expression of the troponin I, α-actin and myoglobin genes. Simultaneously, TAH-11 could be successfully transfected with the GFP reporter gene, suggesting that the TAH-11 cell line could be used for exogenous gene expression in vitro. The TAH-11 cells showed susceptibility to CGSIV, which was demonstrated by the presence of a severe cytopathic effect, suggesting that the TAH-11 cell line is an ideal tool for studying host–virus interaction and potential vaccines. In addition, the expression of inflammatory factors in TAH-11 cells can be remarkably induced following *Streptococcus agalactiae* or *Streptococcus iniae*. The present data lays a foundation to further explore the mechanism of how *Streptococcus* spp. causes tilapia myocarditis.

**Keywords:** *Oreochromis niloticus*; cell line; fish virology; myocardial inflammation

**Key Contribution:** A novel conditionally immortalized Nile tilapia (*Oreochromis niloticus*) heart cell line was established and characterized. The cell line was identified as a myocardial cell line and sensitive to the CGSIV virus, which can be used for developing cell models for fish virology and myocarditis studies.

## 1. Introduction

Tilapia (*O. niloticus*) is a globally important freshwater food fish well suited for aquaculture production [1]. However, the outbreak of various bacteria and viruses has posed a significant threat to the global tilapia-farming industry, especially over the past decade [2,3]. Streptococcosis is one of the most frequently reported bacterial diseases in tilapia aquaculture [4]. The spread of *Streptococcus* has seriously endangered the development of tilapia aquaculture and caused significant economic losses to tilapia aquaculture [5]. Previous studies have shown that *Streptococcus* can cause damage to multiple organs of tilapia, including the brain, liver, heart and kidney [6–8].

Myocardial inflammation can be caused by various factors, such as bacterial infection, cardiotoxic drugs or mechanical damage, with the most common bacterial infection being, *Streptococcus* spp. [9]. *S. agalactiae* or *S. iniae* often causes severe myocarditis in teleost fishes, including tilapia [10–15]. The histological features of myocarditis mostly manifest as edema, inflammatory cell infiltration and myocardial necrosis. Bacterial myocarditis is the key cause of myocardial injury and cardiac insufficiency [16]. In the pathological process of bacterial myocarditis, excessive inflammatory reactions can lead to myocardial cell death and myocardial injury [17].

Fish cell culture technology, as an important research tool, has been widely used in the fields of toxicology, carcinogenesis, gene regulation and expression, DNA replication and repair, etc. [18–21]. In 1962, the first teleost cell line, the rainbow trout (*Oncorhynchus mykiss*) gonadal cell line RTG-2, was reported in the literature [22]. Since then, the research on fish cell culture has shown rapid development. Viral infection has been a major challenge for the aquaculture industry, resulting in high mortality and significant production losses. Cellular models are essential for studying viral pathogenesis and properly monitoring viral diseases. Moreover, compared with animal experiments, cell culture is more efficient, economical and in line with animal welfare.

The present study established and characterized a new cell line derived from tilapia heart. This newly established tilapia cell line was named TAH-11. The cell line expressed various myocardial cell markers, including troponin I, $\alpha$-actin and myoglobin [23–25]. This indicated that the cells were myocardial cells, which could be used as a research tool for tilapia myocarditis. In addition, the cells were sensitive to CGSIV in viral infection experiments. Thus, TAH-11 cells were identified as myocardial cells that could be applied to the research on fish virology and myocarditis studies.

## 2. Materials and Methods

### 2.1. Primary Cell Culture and Routine Maintenance

Healthy Nile tilapia (*O. niloticus*) $10 \pm 1.0$ cm in length was gifted by Prof. De-Shou Wang (School of Life Sciences, Southwest University, Chongqing, China) and cultured at Guangdong Ocean University. These fish had been sourced from cultured populations of Japan (the Laboratory of Professor Yoshitaka Nagahama of Okazaki National Institute of Basic Biology) originating from stocks in Egypt. In the laboratory, tilapia was euthanized with an overdose (200 mg/mL) of tricaine MS-222 (Sigma, Darmstadt, Germany) and wiped with 70% ethanol. The heart of tilapia was removed by anatomical tools treated with autoclaves and immersed in phosphate-buffered saline (PBS) containing 5× antibiotics (400 IU/mL penicillin, 400 μg/mL streptomycin and 2.5 μg/mL amphotericin B) for three minutes. The heart was cleaned 3 times with Leibovitz's L-15 (Gibco, New York, NY, USA) containing 5× antibiotics (400 IU/mL penicillin, 400 μg/mL streptomycin and 2.5 μg/mL amphotericin B). Then, the heart was digested with 1 mL of 0.25% EDTA–Trypsin (Gibco, New York, NY, USA) for 15 min to soften up the heart and minced into small pieces (approximately 1 mm$^3$ in size). After filtration through a 0.45 μm filter membrane, the contents were centrifuged at 400× *g* for 5 min. The supernatant containing 0.25% EDTA–Trypsin was discarded and suspended with L-15 medium supplemented with 20% FBS (Gibco, New York, NY, USA) and 3× antibiotics (400 IU/mL penicillin, 400 μg/mL streptomycin and 2.5 μg/mL amphotericin B) and, respectively, seeded in three 25 cm$^2$ culture flasks (Corning, New York, NY, USA). The flasks were incubated at 25 °C, and half a volume of medium was replaced every three and a half days. Once 95% confluence of primary cells was formed, the cells were subcultured in a new 25 cm$^2$ at the ratio of 1:2.

In each passage, the old culture medium was discarded, and the cells were cleaned with 2 mL of PBS before they were digested with 2 mL of 0.25% trypsin-EDTA solution at 37 °C for 5 min. Then, the supernatant was discarded after centrifuging at 400× *g* for 5 min, suspended with L-15 medium containing 20% fetal bovine serum, transferred to 25 cm$^2$ culture flasks (Corning, New York, NY, USA) and incubated at 28 °C. After 15 passages, the concentration of FBS in the L-15 medium decreased to 10%.

## 2.2. Optimization of Cell Culture Condition

The effect of different media, FBS concentrations and temperatures on cell growth was investigated at Passage 50 of TAH-11 cells in this study. The density of $0.2 \times 10^4$ cells was seeded in 6-well plates with medium (L-15, DMEM, MEM and M199) containing 10% FBS and cultured at 28 °C. To determine the effect of the different FBS concentrations on the proliferation of TAH-11, the cells were seeded by L-15 with concentrations of FBS (5%, 10%, 15% and 20%) and cultured at 25 °C. The effect of cell culture temperature on cell growth was evaluated in the different 6-well plates. The plates were incubated at 18 °C, 25 °C, 32 °C and 39 °C. Cell proliferation was measured every day after inoculation. Cells from three wells under each experimental condition were digested with trypsin and counted with a cell counter. The experiment lasted for 7 days, and the number of cells was expressed as mean $\pm$ SD.

## 2.3. Cryopreservation and Revival of Cell

After 10 passages, TAH-11 cells were cryopreserved in each 5th passage. In brief, cell cultures were suspended in serum-free cryopreservation solution (Beyotime, Shanghai, China) after digestion with trypsin-EDTA solution and centrifuged at $400\times g$ for 5 min. The density of $2 \times 10^6$ was aliquoted into 2 mL sterile microtubules, placed in the freezing container (Corning, New York, NY, USA), stored overnight at $-80$ °C and stored in liquid nitrogen for long-term cryostorage.

To facilitate resuscitation, cell vials from liquid nitrogen were quickly thawed in a 37 °C water bath for 2 min and were dripped into 2 mL of L-15 medium containing 10% FBS. Then, the cells were suspended with 5 mL of fresh complete medium after centrifuging at $400\times g$ for 5 min and seeded in a 25 cm$^2$ cell culture flask at 25 °C. After 24 h, the old medium was replaced by L-15 supplemented with 10% FBS and 1$\times$ antibiotics (400 IU/mL penicillin, 400 µg/mL streptomycin and 2.5 µg/mL amphotericin B).

After the cells were thawed, a small amount of cell suspension was collected into a centrifuge tube, stained with trypan blue and observed under a microscope, and cell viability was calculated.

## 2.4. Chromosome Analysis

TAH-11 cells were seeded in a 75 cm$^2$ culture flask at 25 °C for 24 h at Passage 60. Then, chromosomal analysis was performed following the method described by the manufacturer's instructions of the GENMED blue fluorescent chromosome karyotype analysis kit and Giemsa. The cell cultures grown to 75% of the flask were cleaned by GENMED Reagent A, and 100 µL of preheated GENMED Reagent B was and inoculated at 25 °C for 12 h. The cells were resuspended with 10 mL of preheated GENMED Reagent C at 37 °C for 25 min and fixed in freshly prepared and cold GENMED fixative. The cell suspension was dropped from a height of 50 cm onto a precooled glass slide, avoiding duplicating areas to ensure chromosome distribution. The slides were stained with Giemsa solution (Beyotime, Shanghai, China) for 20 min after air-drying, and the number of chromosomes in 100 cells in the metaphase phase was counted under a light microscope (Zeiss, Jena, Germany).

## 2.5. Identification of TAH-11 and Detection of Mycoplasma Contamination

To identify the origin and type of TAH-11, heteroduplex analysis was performed on the mitochondrial 18S rRNAs of the cells at Passage 80. In addition, we identified the expression of myocardial cell markers in TAH-11 cells, including troponin I, $\alpha$-actin and myoglobin [23–25]. The primers of genes were designed by primer 5.0 (Table 1). After PCR amplification and 1% agarose gel containing GelRed electrophoresis detection, the PCR was sequenced by Sangon Biotech (Shanghai, China). The sequences were aligned against known sequences in the National Centre for Biotechnology Information database using the Basic Local Alignment Search Tool (BLAST) to determine the cell origin of TAH-11.

**Table 1.** Sequences of primers used in this study.

| Primer Name | Sequence (5′–3′) | GenBank Accession | Primer Efficiency |
|---|---|---|---|
| q-*Caspase3*-F | CGAAACGGTACTGACGTGGA | NM_001282894.1 | 96% |
| q-*Caspase3*-R | GAGCCGTCCGTACCAAAGAA | | |
| q-*Caspase9*-F | GTTGTCCGCCCTGTAATCCA | XM_025901776.1 | 93% |
| q-*Caspase9*-R | GTCTTAACTGCCACCCGTCA | | |
| q-*TNF-α*-F | CTCAGAGTCTATGGGAAGCAG | XM_025902124.1 | 97% |
| q-*TNF-α*-R | GCAAACACGCCAAAGAAGGT | | |
| q-*TGF-β*-F | TGGGACTATGAGCAGGAGGG | NM_001311325.1 | 96% |
| q-*TGF-β*-R | AACAGCAGTTGTGTGATTGGGT | | |
| q-*IL-8*-F | GATAAGCAACAGAATCATTGTCAGC | XM_019359413.2 | 97% |
| q-*IL-8*-R | CCTCGCAGTGGGAGTTGG | | |
| q-*β-actin*-F | AACAACCACACACCACACATTTC | XM_003455949.5 | 98% |
| q-*β-actin*-R | TGTCTCCTTCATCGTTCCAGTTT | | |
| *18S rRNA*-F | TTCATTGATGCACGAGCCGA | MF460356.1 | |
| *18S rRNA*-R | CGCCGAGAAGACGATCAAAC | | |
| q-*myoglobin*-F | CCATGGAGCCACTGTGCTAA | NP_001266612.1 | 94% |
| q-*myoglobin*-R | GGGATCTTGTGCTTTGTCGC | | |
| q-*troponin I*-F | TCGAGGTACGACACCGAGAT | XM_025909096.1 | 98% |
| q-*troponin I*-R | TCTTCAGGGCGGGTTTCTTC | | |
| q-*α-actin*-F | TTGAGGCCAGATGAGAAGGC | XM_025901239.1 | 95% |
| q-*α-actin*-R | GACGGATCCACTCCAGCAAA | | |

To determine whether the TAH-11 cell line was not contaminated with mycoplasma, mycoplasma contamination was checked at Passage 60 according to the manufacturer's instructions of TransDetect PCR Mycoplasma Detection Kit (Beyotime, Shanghai, China). The density of $1 \times 10^6$ TAH-11 cells was inoculated in a 25 cm$^2$ culture flask with L-15 containing 10% FBS at 25 °C for 24 h. Harvest 2 mL of the supernatant and centrifuge at $12,000 \times g$ for 30 min when the cell cultures reached 80–90% confluence. The deposit was resuspended with 50 μL of 1× TE buffer and heated at 95 °C for 10 min. A 2.5 μL volume of the supernatant centrifuged again was amplified as a template for PCR and according to 1% agarose gel containing GelRed electrophoresis detection.

### 2.6. Transfection Efficiency Analysis

The transfection efficiency of TAH-11 was investigated at Passage 70 in this study. The green fluorescent protein (GFP) expression vector pGFP-N1 (Beyotime, Shanghai, China) was transfected into TAH-11 cells and 293T cells after the cells were inoculated in 12-well plates (Corning, New York, NY, USA) at 25 °C for 24 h. The monolayer cell was washed with L-15, 1 μg of plasmid was added per well according to the manufacturer's instructions for the transfection reagent Lipofectamine 3000 (Invitrogen, Waltham, MA, USA), and the results of the transfection and negative controls were observed after 48 h under a fluorescence microscope (Zeiss, Jena, Germany).

### 2.7. Virus Susceptibility Analysis

The susceptibility of TAH-11 cells to CGSIV was investigated at Passage 55 in this study. Chinese giant salamander iridovirus (CGSIV) was isolated in China by Yu Dapeng and used under authorization [26]. A 0.5 mL volume of a suspension of $5 \times 10^4$ TAH-11 mL$^{-1}$ in L-15 supplemented with 10% FBS was seeded in 24-well culture plates and cultured at 25 °C. When TAH-11 cells reached 80% confluence, the cell cultures per well were inoculated with 100 μL of CGSIV suspension. The cells were cultured at 25 °C, and the infected cultures were monitored by an inverted microscope in real time to determine the cytopathic effects (CPE).

*2.8. Transmission Electron Microscopic (TEM) Analysis of CGSIV and Streptococcus spp. to TAH-11*

TAH-11 cells were infected with CGSIV (MOI = 2) and *S. agalactiae* (MOI = 10), respectively. *Streptococcus* spp. infected cells were collected by centrifugation at 12 h p.i and CGSIV-infected cells were collected by centrifugation at 48 h p.i. The cells were fixed in 0.1 M PBS (pH 7.4) containing 2.5% glutaraldehyde for 24 h at 4 °C, and then in 0.1 M PBS (pH 7.4) containing 2.0% osmium tetroxide. Then, the cells were dehydrated in fractionated ethanol, embedded in epoxy resin and sliced. Ultrathin sections were stained with uranyl acetate/lead citrate and observed under an electron microscope.

*2.9. Adhesion Analysis of Streptococcus spp. to TAH-11*

The cells at the concentration of $1 \times 10^4$ were added to each of the 24 wells of a microtiter plate and cultured for 48 h at 25 °C. After the cells reached 90% confluence, the L-15 medium was removed, the cell cultures were cleaned with PBS three times, and 0.5 mL of antibiotic-free L-15 medium was added to each of the 24 wells. Then, *S. agalactiae*, *S. iniae* and *E. coli* (DH5α) (final concentration $1 \times 10^5$ CFU/mL) were added to each of the 24 wells, separately. The medium containing bacteria was harvested after co-incubation for 1, 2, 3 and 4 h, and cells were cleaned three times with PBS and lysed with 250 μL 1% Triton-X. The suspension was diluted in a gradient fashion and dropped onto antibiotic-free BHI solid medium to count the live bacteria, and the adhesion rate was calculated. Each adherence experiment was performed in technical triplicate on a single plate, and values were averaged. The results shown are the averages from at least three biological replicates. Data were analyzed by one-way ANOVA with a Kruskal–Wallis post hoc test using GraphPad Prism 9 software.

*2.10. Analysis of Fish Immune-Related Gene Expression with Streptococcus spp.*

The density of $1 \times 10^6$ TAH-11 cells per well was seeded in 6-well culture plates with antibiotic-free L-15 medium and cultured at 25 °C for 24 h. *S. agalactiae* and *S. iniae* (final concentration $1 \times 10^5$ CFU/mL) were added respectively to 6-well culture plates. After 1 h, 2 h, 3 h and 4 h infection, the total RNA of TAH-11 cells from each treatment (*S. agalactiae*/*S. iniae*/no-treatment control) was extracted according to RNAiso Plus (TaKaRa, Dalian, China) and reversely transcribed to cDNA using PrimeScript™ RT reagent kit with gDNA Eraser (TaKaRa, Dalian, China). The SYBR Green I RT-PCR assay was performed on a real-time PCR System (Thermo Fisher Scientific, Waltham, MA, USA) following the manufacturer's instructions. The primers are shown in Table 1. The expression levels of each sample were calculated via the $2^{-\Delta\Delta Ct}$ method [27].

All data were shown as the mean with standard deviation, and the significant difference was analyzed using one-way ANOVA and the Student's *t*-test through the SPSS statistics 17.0. Meanwhile, the statistically significant differences ($p < 0.05$) were revealed by asterisks.

## 3. Results

### 3.1. Primary Cell Culture and Routine Maintenance

TAH-11 cell lines were established from the heart of tilapia fingerlings with a trypsin solution containing EDTA. TAH-11 cells attached to the bottom of 6-well culture plates one to two days after seeding and grew into a monolayer within 7 days at 25 °C. Morphologically, the primary cells consisted of both epithelial-like cells and fibroblast-like cells (Figure 1). Before the 10th passage, the TAH-11 cells were cultured by L-15 containing 20% FBS and 1× antibiotics (400 IU/mL penicillin, 400 μg/mL streptomycin and 2.5 μg/mL amphotericin B), and the passage interval was 5 days at a ratio of 1:1.5 by trypsin solution. From Passages 10 to 20, the TAH-11 cells grew faster and subcultured in a ratio of 1:3, and the time was shortened to 3 days. From Passage 20, the FBS was reduced gradually to 10% in the L-15 culture medium. TAH-11 gradually was stable and became fairly homogenous in size and reached confluency on the second day of culture after Passage 50. In the course

of nine months, the cells were subcultured more than 80 times (Figure 1). After the 30th cell culture passage, the cell morphology gradually showed a single flat epithelioid. At Passage 80, the cultures had become a monoculture of epithelial-like cells. The developed cell line from the heart has been designated as the tilapia heart cell line (TAH-11).

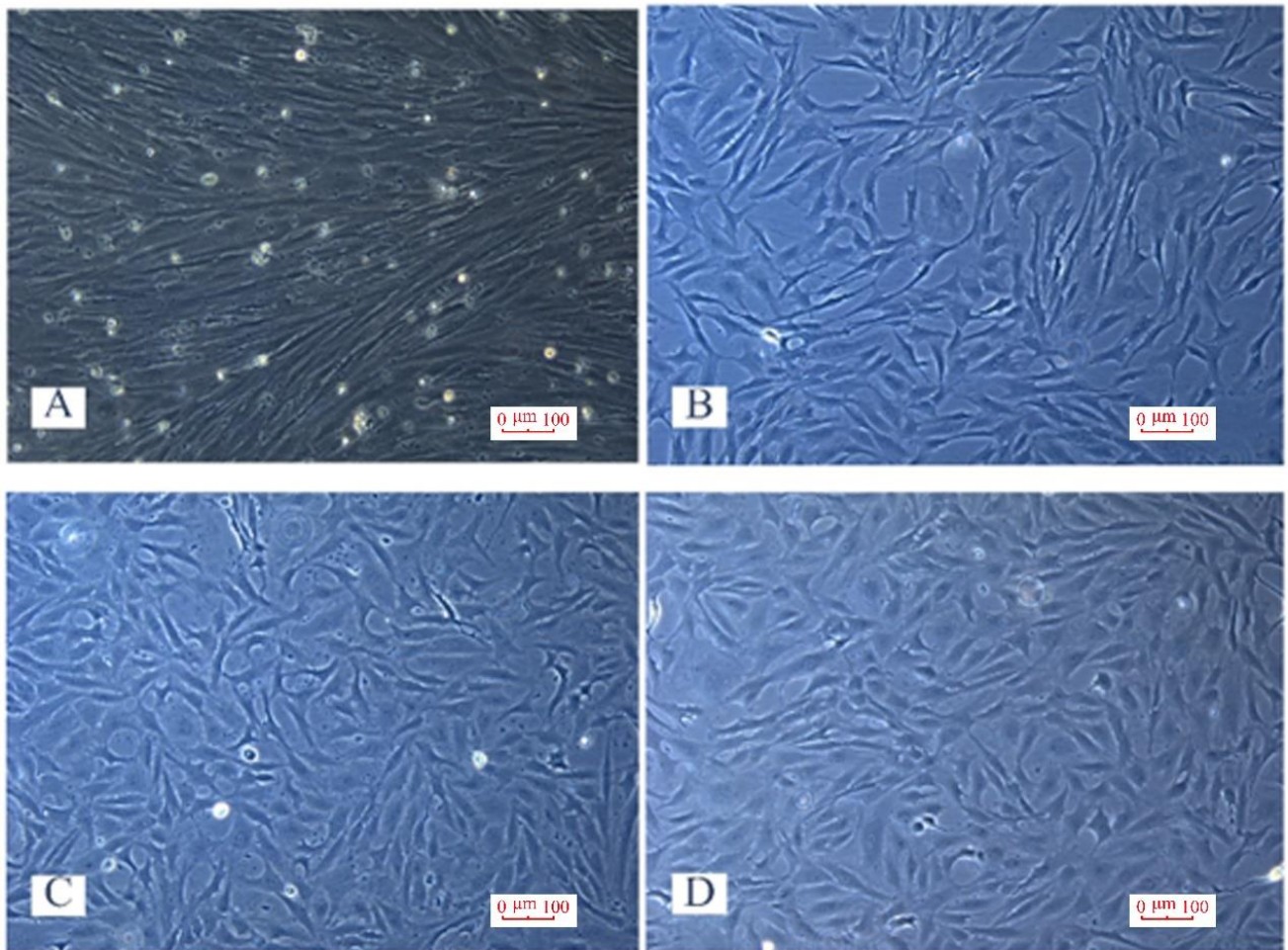

**Figure 1.** Morphology of TAH-11 cell monolayers: (**A**) Passage 5; (**B**) Passage 20; (**C**) Passage 50; (**D**) Passage 70. Scale bar = 100 μm.

### 3.2. Optimization of Cell Culture Condition

As shown in Figure 2, the TAH-11 cell line can grow in L-15, DMEM and MEM medium, but the growth rate of TAH-11 in L-15 is significantly faster than in other mediums at the same serum concentration and culture temperature (Figure 2A). The TAH-11 cells can grow over a wide range of temperatures, and it was optimal at 25 °C (Figure 2C). The TAH-11 cell growth rate increased with the concentration of serum, and the proliferation of cells with 5% FBS was lower compared to others (Figure 2B). In consideration of cost and experimental requirements, we chose L-15 medium containing 10% FBS and 25 °C to culture cells in a further study.

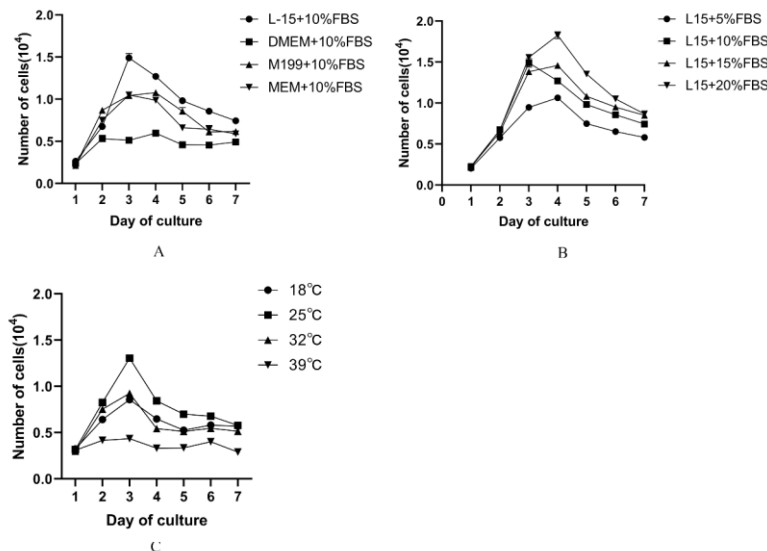

**Figure 2.** Growth kinetics of TAH-11cell line at Passage 50: (**A**) effects on cell growth of different medium at 25 °C; (**B**) effects of different concentrations of FBS in L-15 medium on cell growth at 25 °C; (**C**) effects of different culture temperatures on cell growth in L-15 medium containing 10% FBS.

### 3.3. Cryopreservation and Revival of Cell

The TAH-11 cells were cryopreserved at each 5th passage and showed 90% viability after recovering from liquid nitrogen. The recovered cells were fully confluent after 48 h, and the morphology of cells was stable after freezing and thawing.

### 3.4. Chromosome Analysis

The metaphase mitotic figures were prepared on the chromosomes of TAH-11 cells, and karyotype analysis was performed (Figure 3). In 100 mitoses, the number of chromosomes is between 26 and 50, but the maximum number of chromosomes in mitosis is 42, which indicates that the number of chromosomes in the cell should be 42.

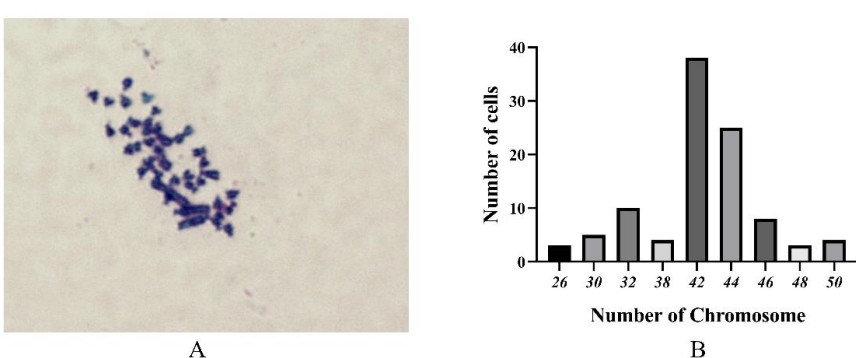

**Figure 3.** Chromosome morphological features and frequency distribution of TAH-11-11 cells: (**A**) chromosome morphology of TAH-11 cells at Passage 50; (**B**) chromosome numbers were counted in colchicine-treated TAH-11 cells at Passage 50.

### 3.5. Identification of TAH-11 and Detection of Mycoplasma Contamination

To identify the molecular biology of the TAH-11 cell lines, we referred to a previous similar study [28]. Heteroduplex gene analysis was used for the molecular biological identification of the immortalized TAH-11 cell line. The results showed that the expected 691 bp fragment of partial 18S rRNA, the 115 bp fragment of the β-actin 10⁶ bp fragment of troponin I, the 184 bp fragment of α-actin and the 112 bp fragment of myoglobin were obtained. The amplified fragments were sequenced and compared with 18S rRNA of *O. niloticus* in NCBI (GenBank Accession No. MF460356.1), and the comparison of results

showed that the homology of the two sequences was over 99%, suggesting that the TAH-11 cell line originated from *O. niloticus* (Figure 4). The partial sequence result of the 18S rRNA gene obtained in this study is shown in Figure 4B.

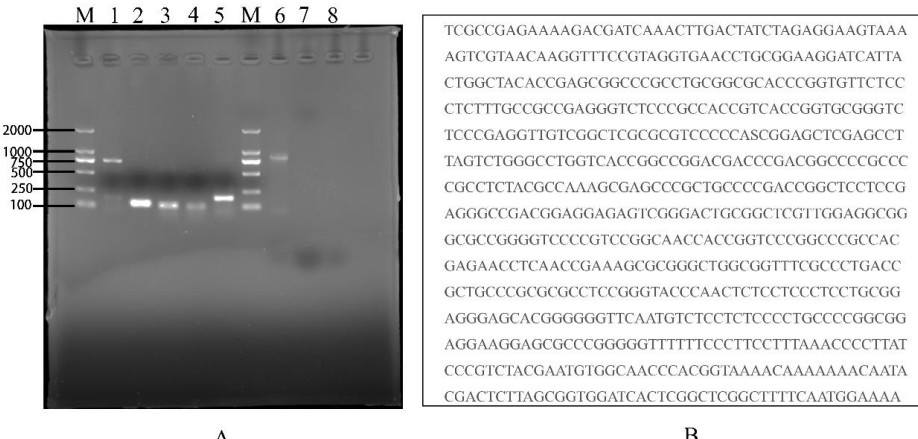

A

B

**Figure 4.** Multi-gene identification and detection of mycoplasma contamination. 18S rRNA amplification and mycoplasma testing (**A**) M: DL2000 DNA marker; Lanes 1: 18S rRNA (691 bp); Lanes 2: β-actin (115 bp); Lanes 3: myoglobin (112 bp); Lanes 4: troponin I (106 bp); Lanes 5: α-actin (184 bp); Lanes 6: Positive control; Lanes 7: Negative control; Lanes 8: TAH-11 cell supernatant; (**B**) sequencing results of the 18S rRNA amplification.

The mycoplasma contamination test demonstrated that no amplicon was found in the amplification results using TAH-11 cells as templates, indicating that the TAH-11 cell line was not contaminated with mycoplasma (Figure 4).

### 3.6. Transfection Efficiency Analysis

To investigate transfection efficiency, we transfected the pGFP-N1 plasmid to TAH-11 at Passage 70 and 293T cells. After being transfected with pGFP-N1 for 48 h, the bright-green fluorescence signals were obviously detected in TAH-11 cells and 293T cells (Figure 5). The bright-green fluorescent signals were both detected. The results showed that the TAH-11 cell line has the potential to study foreign genes.

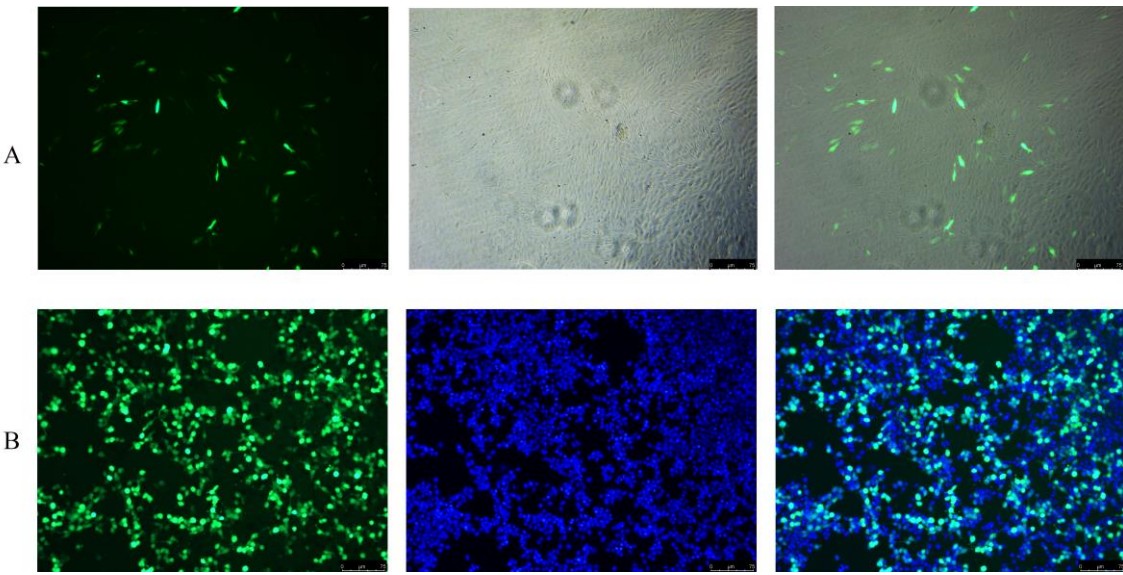

**Figure 5.** Expression of GFP reporter gene in TAH-11 cells and 293T cells: (**A**) TAH-11; (**B**) 293T cells. Scale bar = 75 μm.

### 3.7. Virus Susceptibility Analysis

The susceptibility of the TAH-11 cell line to CGSIV was determined by observing CPE under a fluorescence microscope. Microscopic observation showed that the CGSIV-infected cells became shrunken, round and obvious voids in the cell monolayer (Figure 6). With the prolongation of the infection time, the TAH-11 cells completely lysed and began to detach from the culture flask, and the whole-cell monolayer gradually formed a reticular connection. In conclusion, all data indicated that CGVIS can infect and cause cytopathic effects in TAH-11 cells. This showed that the TAH-11 cell line can be used to investigate the pathogenesis of CGSIV in vitro.

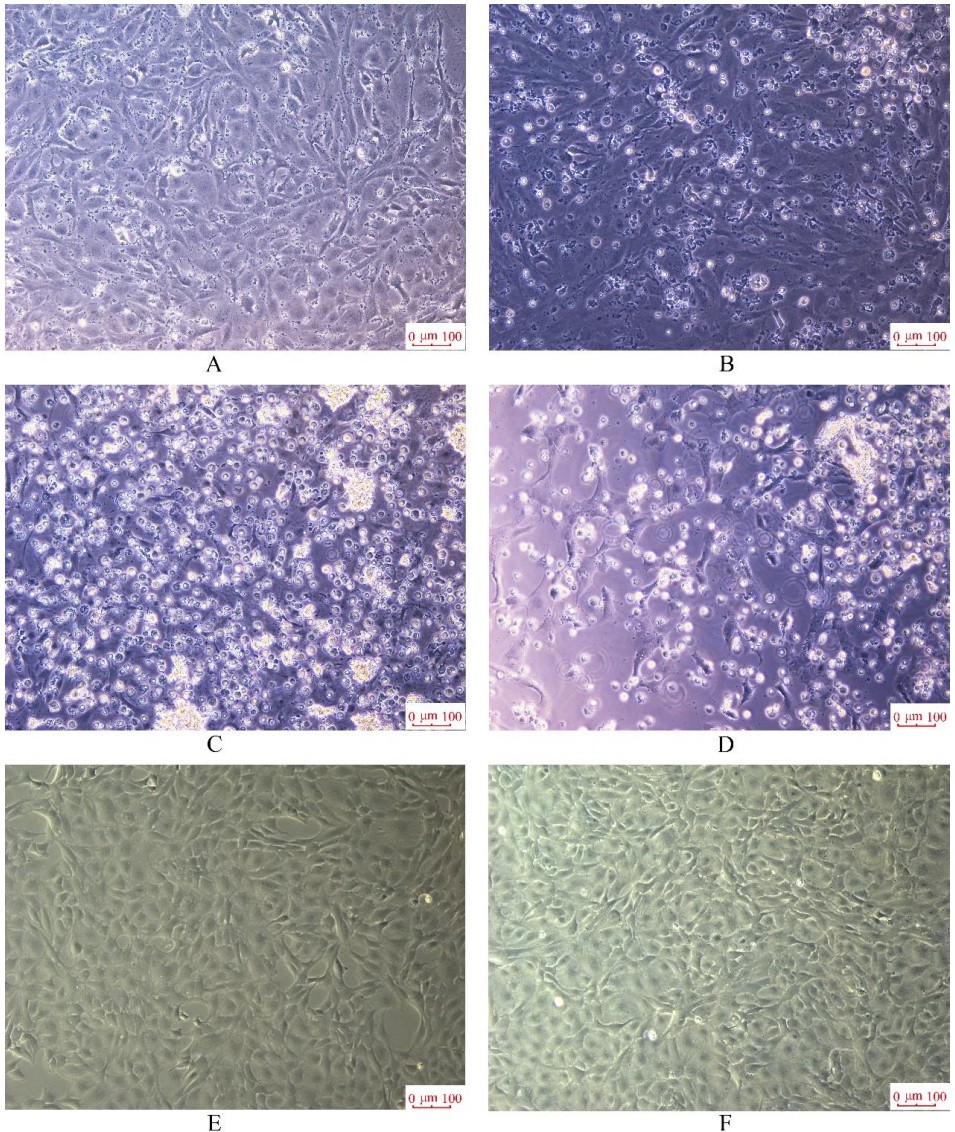

**Figure 6.** Replication characteristic of CGSIV in TAH-11 cells: (**A**) CPE of TAH-11 cells infected with CGSIV at 12 h p.i; (**B**) CPE of TAH-11 cells infected with CGSIV at 24 h p.i; (**C**) CPE of TAH-11 cells infected with CGSIV at 36 h p.i; (**D**) CPE of TAH-11 cells infected with CGSIV at 48 h p.i; (**E**) L-15 with 10% FBS treatment; (**F**) L-15 without 10% FBS treatment. Scale bar = 100 μm.

### 3.8. Transmission Electron Microscopic (TEM) Analysis of CGSIV and Streptococcus spp. to TAH-11

The morphology of CGSIV and *Streptococcus* spp. in TAH-11 cells was further observed by transmission electron microscope. A large number of viral particles were observed in the cytoplasm of the TAH-11 cells infected with CGSIV (Figure 7A). The viral particles are

hexagonal in shape and have a diameter of about 100 to 120 nm. This showed that CGVIS can infect and cause cytopathic effects in TAH-11 cells. In addition, in the experiment of *Streptococcus* spp. infecting TAH-11 cells, *S. agalactiae* was observed inside the cell (Figure 7B), which also showed that the heart of tilapia was a suitable organ for *S. agalactiae* to replicate.

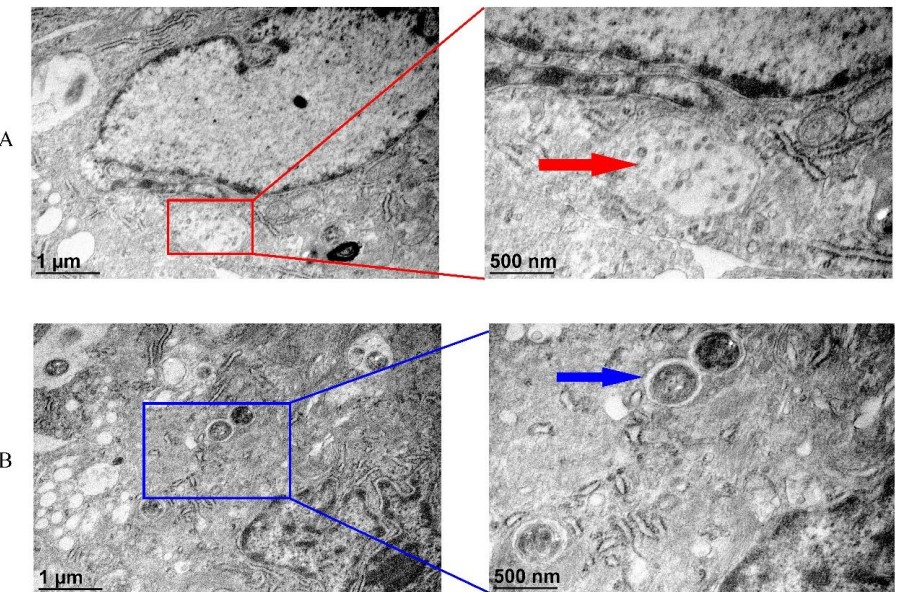

**Figure 7.** Transmission electron micrograph of TA-11 infected by CGSIV: (**A**) areas indicated by red arrows containing virus particles; (**B**) transmission electron micrograph of TA-11-infected *S. agalactiae*. The areas indicated by blue arrows contain bacteria.

*3.9. Adhesion Analysis of Streptococcus* spp. *to TAH-11*

In order to evaluate the adhesion of *S. agalactiae* and *S. iniae* to TAH-11 cells, *E. coli* (DH5$\alpha$) was considered to be a nonpathogenic bacterium and served as a control [29]. As shown in Figure 8, analyzing the adherence rates of three strains, we observed that the adhesion of *S. agalactiae* and *S. iniae* to TAH-11 cells increased with time, but the adhesion of *E. coli* (DH5$\alpha$) increased to a smaller extent. These results indicated that *Streptococcus* spp. can adhere to TAH-11 cells, which provided a basis for *S. agalactiae* and *S. iniae* to infect TAH-11 cells.

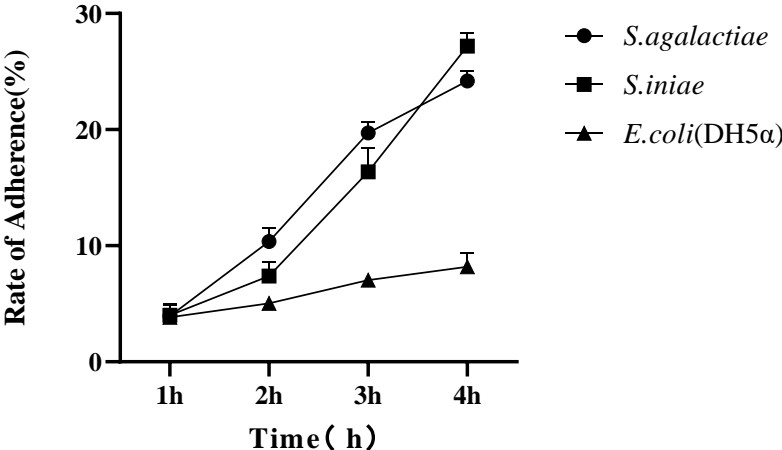

**Figure 8.** Adherence assay of *S. agalactiae*, *S. iniae* and *E. coli* (DH5$\alpha$) to TAH-11 cells.

### 3.10. Analysis of Fish Immune-Related Gene Expression

In order to explore the host immune response to *Streptococcus* spp., we analyzed the effects of *S. agalactiae* and *S. iniae* on the expression of inflammatory factors in TAH-11 cells by qRT-PCR (Figure 9). The results showed that the expression of the proinflammatory factor gene *TNF-α* was significantly increased ($p < 0.05$) in the infected group, and the expression of the anti-inflammatory factor gene *TGF-β* was downregulated ($p < 0.05$) compared with that in the PBS group after co-incubation for 1 h. However, after 3 h, *TNF-α* was downregulated, while *TGF-β* was significantly upregulated ($p < 0.05$). The mRNA expression levels of *TNF-α* showed a significant up-regulation ($p < 0.05$) after *S. iniae* infection at 3 h, while the expression of *TGF-β* increased at 2 h. *Caspase3* and *Caspase9* were investigated to evaluate the effect of *S. agalactiae* or *S. iniae* on apoptosis of TAH-11 cells. In comparison to the PBS group, the transcriptional levels of *Caspase3* were significantly increased ($p < 0.05$) after the *S. agalactiae* challenged in the *S. agalactiae* group. From 1 h to 2 h post-challenge, the transcriptional levels of *Caspase3* and *Caspase9* in the *S. iniae* group were markedly upregulated ($p < 0.05$) in comparison to the PBS group. Moreover, the infection of *S. agalactiae* and *S. iniae* also caused the expression of *IL-8* significantly to be significantly higher than that in the control group ($p < 0.05$).

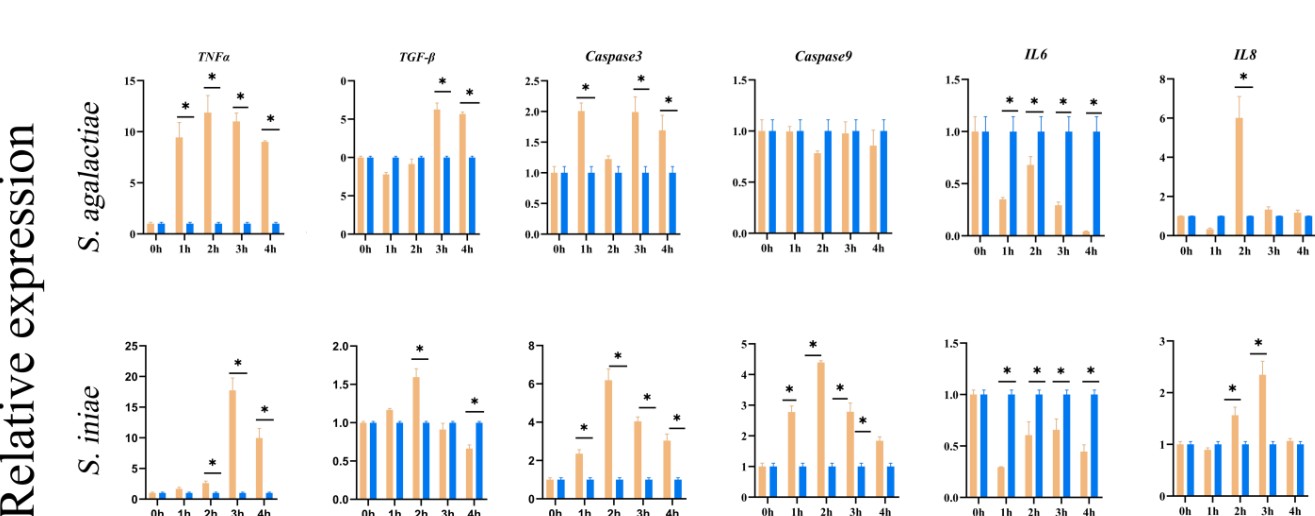

**Figure 9.** The expression patterns of immune-related gene expression (*TNF-α*, *TGF-β*, *Caspase3*, *Caspase9*, *IL-8*) at various points after *S. agalactiae* and *S. iniae* infection via qRT-PCR. Every value is presented as the mean and standard deviation; $n = 3$. The asterisks reveal a significant difference ($p < 0.05$).

## 4. Discussion

Fish cells play an important role in the isolation and identification of fish viruses, functional gene analysis and the preparation of biological products. The first permanent fish cell line RTG-2 was established by Wolf in 1962. Since then, more and more fish cell lines have been established because of the rapid development of fish cell culture technology [30]. The primary cell culture methods are gradually increasing, roughly divided into enzyme digestion, tissue block fixation, mechanical dispersion and complexing agent dispersion. This study combined enzymatic digestion and mechanical separation to separate the heart cells from tilapia for harvesting more heart cells. The TAH-11 cell line has been subcultured for over 80 passages since it was isolated from the heart. The result demonstrates that the TAH-11 cell line can proliferate in vitro.

During the initial passages, the cells have diverse morphologies. However, after multiple separations and purification, the cell morphology tends to be uniform and stable. At Passage 80, the cultures had become a monoculture of epithelial-like cells. In addition, the cells are in good condition after cryopreservation for six months. These results showed

that we established a cell line that can stably propagate in vitro and serve as an excellent experimental tool for toxicological, genetic and immunological research.

In this study, we investigated the optimal growth conditions for the TAH-11 cell line. The results showed TAH-11 cells could grow at a temperature of 16 °C to 37 °C, and the temperature conditions are consistent with tilapia. This indicates that the environmental temperature adapted for cell culture was closely related to the appropriate temperature for live fish. For this reason, when a cell line is established, we should choose the proper growth temperature for this fish to culture it.

Generally, the serum is essential for the proliferation of cells in vitro. It contains various nutritional factors for cell growth. We added 20% FBS to the primary cell culture system to maintain the stable development of cells. In contrast, after the cells were gradually stabilized, the concentration of FBS in the system was gradually reduced to 10%, and cells could rapidly proliferate at this concentration. In the FBS concentration experiment, we found that 10%, 15% and 20% had no significant effect on cell growth, except that 5% FBS was not conducive to cell proliferation. Therefore, considering the cost and experimental needs, we chose to culture cells with 10% FBS. L-15 medium that enriches the amino acid component adopts galactose as a sugar source, and is used for culturing fish cells [31]. L-15 medium can improve the buffering capacity by a high concentration of amino acids, instead of the traditional sodium bicarbonate buffering system. In addition, the substitution of galactose and sodium pyruvate for glucose can prevent the formation of acidic metabolic byproducts and help to maintain the stability of the PH value of the medium. It is suitable for cell culture in an environment free of carbon dioxide. It was found in our experiment that L-15 medium was indeed more suitable for the culture of TAH-11 cells.

Chromosome analysis and species identification are important parts of describing cell lines, which can analyze cell lines' genetic stability and origin. In this study, the number of chromosomes in the 60th-generation TAH-11 cells was 42 in most cases, and 44 in part. This is different from the reported number of 44 chromosomes in tilapia [32], indicating that fish cell lines from the same species are not always composed of the same number of chromosomes, which may result from cell immortalization. In the future, we will pay close attention to the changing trend of the TAH-11 cell chromosome. Sequence alignment of the mitochondrial 18S rRNA gene has been used as a reliable molecular method to accurately identify cell sources of many fishes [33,34]. The expression of 18S RNA and β-actin verified that TAH-11 indeed originated from tilapia. Moreover, we found that TAH-11 cells also express multiple marker genes of myocardial cells, including troponin I, α-actin and myoglobin. Therefore, we considered the TAH-11 cell line to be a myocardial cell line derived from the tilapia heart and undergoing immortalization.

The transfection efficiency of fish cell lines is generally lower than that of mammalian cells, which dramatically limits the study of the gene function of fish cell lines [35]. The traditional methods for introducing foreign DNA into cells include liposome transfection, virus infection transduction and electroporation. However, virus infection transduction and electroporation methods have significant limitations in the use process, so researchers usually use liposome transfection to transfect foreign DNA. The transfection reagents on the market are primarily suitable for transfection in mammals; for this reason, the transfection efficiency of fish cells is faulty. The transfection efficiency of TAH-11 cells was still low compared to 293T cells. Therefore, improving the transfection efficiency of TAH-11 cells will be the goal of our future work.

Virology plays an important role in the development of fish cell lines, which is the original application of cell lines [36]. The outbreak of viral diseases is likely to cause large-scale death of aquatic animals, causing serious economic losses and greatly impeding aquaculture development. Iridoviruses is a kind of broad-spectrum infectious DNA virus, which can infect hundreds of hosts and is a common pathogen in aquaculture [37]. The establishment of the fish cell culture technology solves the problem of fish virus isolation and detection [38]. In this study, we investigated the susceptibility of the TAH-11 cell line to CGSIV. The typical CPE and virus particles were observed in TAH-11 cells infected with

CGSIV. This showed that TAH-11 cells were sensitive to the test viruses. This may imply that tilapia is another potential host of Iridoviruses in freshwater. In addition, the TAH-11 cell line can be considered a model suitable for infection studies with CGSIV in vitro, and it is ideal for diagnostic and vaccine research for the benefit of the aquaculture industry.

Bacterial myocarditis is a heart disease caused by infection of bacteria or toxins of bacteria on myocardial tissue. Both *S. agalactiae* and *S. iniae* can cause systemic sepsis and pathological changes in various organs and tissues of tilapia [39]. Myocardial myofibrils of tilapia infected will be broken and necrotic, and the number of mitochondria in the myofibril gap will decrease [40]. *Streptococcus* spp. causes heart infection by gaining access to the bloodstream, adhering to host extracellular matrix proteins, and colonizing myocardium [41]. *Streptococcus* spp. adherence to host cells is an important preliminary step for successful colonization. To ascertain the pattern of colonization of cardiomyocytes by *Streptococcus* spp., we examined the adhesion of *S. agalactiae* or *S. iniae* to TAH-11 cells. The results showed that the adhesion rate of *Streptococcus* spp. to TAH-11 cells increased with the prolongation of infection time. *Streptococcus* spp. infection will initiate an inflammatory cascade in the heart [42]. A variety of key molecules, cytokines and chemokines closely control the inflammatory period. The excessive inflammatory response will adversely affect the damaged myocardium, resulting in myocardial cell death and fibrosis [17]. In the pathological process of bacterial myocarditis, excessive inflammatory reactions can cause myocardial cell death and myocardial damage [17]. In this study, we detected the transcriptional levels of *TNF-α* and *TGF-β*(pro-/anti-inflammatory), *Caspase3* and *Caspase9* (apoptosis-related factors), and *IL-8* via qRT-PCR. Consistent with a previous study, our results showed that the transcription levels of proinflammatory factors in TAH-11 cells increased significantly and remained at a high level for a long time after *S. agalactiae* or *S. iniae* challenged. This might be the cause of the excessive inflammatory reactions of bacterial myocarditis. How bacterial myocarditis damages the myocardial cells of tilapia is still unknown, and further research is needed. Therefore, we evaluated the role of two apoptosis-related factors in the response of TAH-11 cells to streptococcal infection.

Apoptosis is physiological suicide behavior regulated by genes, which is mainly triggered by exogenous (death receptor pathway) or endogenous (mitochondrial pathway) signaling pathways, and then the apoptosis protease is activated to change the cell morphology [43]. In mammals, apoptosis has been proposed to be an essential mechanism for myocardial injury [44,45]. In this study, the results of in vitro experiments showed that the expressions of apoptosis-related factors were upregulated in TAH-11 cells after *S. agalactiae* or *S. iniae* challenged, suggesting that the myocardial cells of Nile tilapia could be induced to apoptosis in bacterial myocarditis. Caspases are distributed in the cytoplasm and play an important role in mediating apoptosis. Under normal conditions, caspase maintains a state of the nonfunctional zymogen. When stimulated by apoptosis-inducing factors, the zymogen is activated into an active caspase for apoptosis. At present, studies show that Caspase3 and Caspase9 are the most closely related to mitochondria. It was generally believed that there are two main ways to activate Caspase3 and Caspase9: Cyt C was released from the mitochondria to the cytoplasm, which combines with Apaf-1 and Caspase9 proenzyme to form an apoptotic body and activate Caspase3. In addition, When the mitochondrial transmembrane potential decreased, not only Cyt C but also apoptosis-inducing factor AIF was released, and AIF could directly activate Caspase3 and then Caspase9 in the cytoplasm [46]. In our experimental results, apoptosis-related factors (caspases 3 and 9) were upregulated after both *S. agalactiae* and *S. iniae* exposure, but biologically speaking, *S. agalactiae* did not have an effect, whereas *S. iniae* did. This may indicate that they have different pathogenic mechanisms. In the *S. agalactiae* infection group, *Caspase3* was only marginally upregulated (2-fold), and *Caspase9* was not altered at all. This was in contrast to 6- and 4-fold induction in the Streptococcus iniae group. Do these results indicate that although all of them belong to the *Streptococcus* spp. and can cause bacterial myocarditis, the detailed infection mechanism is different? This all needs our further research. *IL-8*, also known as the chemokine CXCL 8, can cause a large number of inflammatory factors to

transfer to the infected site, leading to the infiltration of local inflammatory cells and the expansion of the inflammatory response. In this study, the expression level of *IL-8* in the infected group was significantly higher than that in the control group, which indicated that *S. agalactiae* or *S. iniae* infection could lead to an increase of *IL-8* level in the heart of tilapia and induce excessive myocardial inflammation. When *Streptococcus* spp. infects tilapia, the heart is often easily ignored by researchers. However, clinical data indicate that the heart is a target organ for *Streptococcus* spp. to attack tilapia [12,47,48]. In addition, as the circulation center of the body, the heart is closely related to the life and health of the body. Therefore, using the TAH-11 cell line as a research tool for tilapia heart disease will surely make an important research contribution to the prevention and treatment of tilapia disease.

## 5. Conclusions

In summary, we established a cell line named TAH-11 from the heart of the Nile tilapia (*O. niloticus*), which can stably propagate in vitro and serve as a good experimental tool. The cell line exhibits the expressions of myocardial cell markers and supports the replication of CGSIV. Therefore, the TAH-11 cell line can be used for developing cell models for fish virology and myocarditis studies.

**Author Contributions:** Conceptualization, Y.C. and B.W.; methodology, Y.C., Y.L. and B.W.; resources, D.J., D.Z. and J.J.; writing—review and editing, Y.H., J.C. and B.W.; All authors have read and agreed to the published version of the manuscript.

**Funding:** This study was supported by the National Natural Science Foundation of China (Grant No. 32073006, 32002426), the Natural Science Foundation of Guangdong Province (No. 2022A1515010553) and the Guangdong Basic and Applied Basic Research Foundation (2021A1515010956).

**Institutional Review Board Statement:** This study was performed in line with the principles of the Declaration of Helsinki. Approval was granted by the Ethics Committee of Guangdong Ocean University (Date: 10 May 2019).

**Data Availability Statement:** Data are contained within the article.

**Acknowledgments:** Thanks for the support of the Guangdong Provincial Key Laboratory of Aquatic Animal Disease Control and Healthy Culture & Key Laboratory of Control for Diseases of Aquatic Economic Animals of Guangdong Higher Education Institutes.

**Conflicts of Interest:** The authors declare no conflict of interest.

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
