# Peer review of "A New Conditionally Immortalized Nile Tilapia (Oreochromis niloticus) Heart Cell Line: Establishment and Functional Characterization as a Promising Tool for Tilapia Myocarditis Studies"

_fishes, doi:10.3390/fishes8030167_

Round 1
Reviewer 1 Report
Manuscript fishes-2287422 describes the establishment of a heart cell line from tilapia, a globally important warmwater food fish. The techniques presented for establishment and testing of general cell line characteristics appear sound, although description of some of the methods can be improved. There is likely strong future utility of this cell line in tilapia and/or fish myocardial research. Although the manuscript appears scientifically sound, I believe there are a few sections of the Introduction, Methods and Discussion that can and should be improved. My specific comments to improve presentation are below. Note that there are several other typographical and English word choice errors throughout the manuscript that I have not addressed. I suggest the authors work with the handling editors or their own grammatical expert to address this.
Introduction
Ln39-41: I think this is not a particularly strong sentence to start off the manuscript. I would suggest the following alternative “Tilapia (O. niloticus) are a globally important freshwater food fish well suited for aquaculture production [1].”
Ln 65-69: Rather than summarizing the findings, I would suggest changing this to present the aims/objectives of the study. It is important to have the objectives unambiguously laid out in the introduction so readers know what the research goals and study design is aiming to do. Currently this is not presented.
Materials and Methods
Ln74-76: Change this sentence to read “These fish had been sources from cultured populations of Japan (…) originating from stocks in Egypt” for better English sentence structure.
Ln78: “isolated” should be “removed”.
Ln81: Here and throughout the manuscript the word “tissue” is used incorrectly. Tissues are a collection of similar cells that perform a single function (e.g., epithelial, connective, nervous, ect.). You removed the heart, which is an organ consisting of multiple tissue types. So instead of saying “The tissues…” say “The heart…”. Similarly on Ln 83, delete the word tissue and simply say “heart”.
Ln99-108: How was it determined that one temperature, FBS concentration or media type is better than other? If this is done scientifically, the experiment is run with replicates and statistical analysis is applied. It is described that six-well plates were cultured, but the number of replicate wells assessed at each time point need to be stated. Also, in order to say that one temperature produced better growth than another at any given time point some form of statistical probability analysis needs to be applied. In this instance something like an ANOVA would be appropriate. These statistical methods need to be described.
Ln112-114: More detail needs to be provided here. Usually, cell suspensions are frozen at a slow specific rate in an alcohol bath. Was this done or were the vials simply placed on a shelf in a -80 freezer?
Ln115-120: In the results section it is stated that recovered cells had 90% viability. How was this determined? The methods for assessing viability of cryopreserved cells should be described here.
Ln128-129: I do not understand this sentence and suggest amending it for clarification. For chromosome analysis, suspensions of cells are often dropped from a significant height onto a slide to “splat” the chromosomes into a more uniform and non-overlapping field on the slide. Is this what was done here, or did the authors simply air-dry the slides using a fan positioned at 50cm above the slides? It is not clear from the current sentence.
Ln130: How many chromosome clusters were counted? From figure 3 it appears that at least 100 cells were counted. This should be stated.
Ln187: What statistical analysis was performed that compared the different treatment groups? In the results it is stated that the S. iniae was higher than E. coli at 4h, but I don’t see any statistical analysis to support that statement. Additionally, E. coli is shown on Fig 8 graph, but I see no description of its use here in the methods. Please include it.
Ln200-202: I’m glad to see some statistical methods presented, but SPSS has a number of statistical tests that can be run. Which were used here, ANOVA, T-test, non-parametric...? Also note that for gene expression analysis, a log transform is typically applied to the data to ensure normality if using a standard parametic test. Please indicate if this was done.
Results
Ln219: On Ln210-211 it is stated that early cultures appeared to be a mix of epithelial-like and fibroblastic-like cells. Was this still a perceived mix at passage 80 or had the cultures become a monoculture of either fibroblastic- or epithelial-like cells?
Ln230: “…were not significant”. I see no statistical analysis to indicate a probability cutoff that is deemed “significant”. Please provide statistical support for this statement.
Ln262: I do not understand Fig 4B. What is the sequence presented. The legend states “the sequencing results of the products”, but I see six products and 15 lines of sequence with no indication for what is what. This needs to be clarified.
Ln273: I’m not sure what word the authors are looking for, but I’m pretty sure “handsome” is not it. Please clarify.
Ln284: I agree that the EM and standard microscope images show that CGVIS can infect and cause cytopathic effect in this cell line, but this study provides no evidence that CGVIS replicates in this cell line. To make that statement, a titre or viral genome quantity would need to be taken at time of infection and then again later to show that titre and/or genome count had increased. Unless this information is provided, please delete or change this sentence for specific accuracy.
Ln311: the sentence states “significantly higher…” yet I see no statistical support for this statement. Please provide the statistical support. Same with saying the S. iniae is higher than S. agalactiae.
Discussion
Ln349: start a new paragraph with “During the initial passages…”
Ln351: would be good to mention what cell type the uniformity has taken on (e.g., figroblast-like?).
Ln360: Start a new paragraph with “Generally, the serum is..”
Ln368: The discussion regarding different types of media should be expanded. The once sentence here is not enough in my opinion.
Ln373: Specify what the number of chromosomes typically is for Tilapia. This can be chased up from the reference but should be stated in the text as well for easy understanding.
Ln384: Please expand on the differences typically observed between mammalian cell transfection rates and fish, specifically how your transfection rate in this study compares to other mammals and fish.
Ln397: Here would be a good place to explain why CGSIV was specifically chosen to infect this cell line.
Ln401: Although not necessary as a textual addition, I am curious if TAH-11 has been exposed to TLV and if the cell line can culture that virus as well. Given its high global economic impact, if the authors have any information they would like to share this would be a good place to do it.
Ln427: start a new paragraph with “Apoptosis is physiological…”
Ln433: It is stated that apoptosis related factors (caspases 3 & 9) were upregulated after both S. agalactiae and S. iniae exposure, but I would argue that biologically speaking S. agalactiae did not have an effect whereas S. iniae did. Caspase 3 was only marginally upregulated (2-fold) which would have questionable biological relevance and Caspase 9 was not altered at al. Contrast this with 6- and 4-fold respective inductions in response to S. iniae which seem slightly more convincing, although still nothing drastic. I think there needs to be some discussion concerning this point, as well as how it relates to other inflammatory heart responses in fish where 20-50 fold inductions are not be uncommon.
Author Response
Reviewer #1:
- Ln39-41: I think this is not a particularly strong sentence to start off the manuscript. I would suggest the following alternative “Tilapia ( niloticus) are a globally important freshwater food fish well suited for aquaculture production [1].”
Answer: Thanks for your advice. We have revised it according to your advice.
- Ln 65-69: Rather than summarizing the findings, I would suggest changing this to present the aims/objectives of the study. It is important to have the objectives unambiguously laid out in the introduction so readers know what the research goals and study design is aiming to do. Currently this is not presented.
Answer: Thanks for your advice. We have changed summarizing the findings to present the aims of the study on Ln 64-70.
- Ln74-76: Change this sentence to read “These fish had been sources from cultured populations of Japan (…) originating from stocks in Egypt” for better English sentence structure.
Answer: Thanks for your advice. We have revised it on Ln75-77 according to your advice.
- Ln78: “isolated” should be “removed”.
Answer: Thanks for your advice. We have revised it on Ln79 according to your advice.
- Ln81: Here and throughout the manuscript the word “tissue” is used incorrectly. Tissues are a collection of similar cells that perform a single function (e.g., epithelial, connective, nervous, ect.). You removed the heart, which is an organ consisting of multiple tissue types. So instead of saying “The tissues…” say “The heart…”. Similarly on Ln 83, delete the word tissue and simply say “heart”.
Answer: Thanks for your advice. We have revised it according to your advice.
- Ln99-108: How was it determined that one temperature, FBS concentration or media type is better than other? If this is done scientifically, the experiment is run with replicates and statistical analysis is applied. It is described that six-well plates were cultured, but the number of replicate wells assessed at each time point need to be stated. Also, in order to say that one temperature produced better growth than another at any given time point some form of statistical probability analysis needs to be applied. In this instance something like an ANOVA would be appropriate. These statistical methods need to be described.
Answer: Thanks for your advice. We have added the description of the statistical method on Ln108-111 according to your advice.
- Ln112-114: More detail needs to be provided here. Usually, cell suspensions are frozen at a slow specific rate in an alcohol bath. Was this done or were the vials simply placed on a shelf in a -80 freezer?
Answer: Thanks for your advice. The freezing container is required for the vials to be placed in the -80 freezer. We have added the detail on Ln115-118.
- Ln115-120: In the results section it is stated that recovered cells had 90% viability. How was this determined? The methods for assessing viability of cryopreserved cells should be described here.
Answer: Thanks for your advice. After the cells were thawed, a small amount of cell suspension was collected into a centrifuge tube, stained with trypan blue, observed under the microscope, and cell viability was calculated. We have added the description of the methods for assessing viability of cryopreserved cells on Ln125-127.
- Ln128-129: I do not understand this sentence and suggest amending it for clarification. For chromosome analysis, suspensions of cells are often dropped from a significant height onto a slide to “splat” the chromosomes into a more uniform and non-overlapping field on the slide. Is this what was done here, or did the authors simply air-dry the slides using a fan positioned at 50cm above the slides? It is not clear from the current sentence.
Answer: Thanks for your advice. The cell suspension was dropped from a height of 50 cm onto pre-cooled slides and avoided dropping to the same area and the slides were stained after air-dried. We have added the description of chromosome analysis on Ln135-140.
- Ln130: How many chromosome clusters were counted? From figure 3 it appears that at least 100 cells were counted. This should be stated.
Answer: Thanks for your advice. We have added the description of the number of chromosome clusters on Ln139.
- Ln187: What statistical analysis was performed that compared the different treatment groups? In the results it is stated that the iniae was higher than E. coli at 4h, but I don’t see any statistical analysis to support that statement. Additionally, E. coli is shown on Fig 8 graph, but I see no description of its use here in the methods. Please include it.
Answer: Thanks for your advice. We have revised it according to your advice and added the description of the use of E. coli in the methods on Ln191.
- Ln200-202: I’m glad to see some statistical methods presented, but SPSS has a number of statistical tests that can be run. Which were used here, ANOVA, T-test, non-parametric...? Also note that for gene expression analysis, a log transform is typically applied to the data to ensure normality if using a standard parametic test. Please indicate if this was done.
Answer: Thanks for your advice. All data were shown as the mean with standard deviation, and the significant difference was analyzed using the one-way ANOVA and Student’s t-test through the SPSS statistics 17.0. Meanwhile, the statistically significant differences (p < 0.05) were revealed by asterisks.
- Ln219: On Ln210-211 it is stated that early cultures appeared to be a mix of epithelial-like and fibroblastic-like cells. Was this still a perceived mix at passage 80 or had the cultures become a monoculture of either fibroblastic- or epithelial-like cells?
Answer: Thanks for your advice. After the 30th cell culture passage, the cell morphology gradually showed a single flat epithelioid. At passage 80 the cultures had become a monoculture of epithelial-like cells. We have added the description of the cell morphology on Ln232-234.
- Ln230: “…were not significant”. I see no statistical analysis to indicate a probability cutoff that is deemed “significant”. Please provide statistical support for this statement.
Answer: Thanks for your advice. We have revised it on Ln108-111 according to your advice.
- Ln262: I do not understand Fig 4B. What is the sequence presented. The legend states “the sequencing results of the products”, but I see six products and 15 lines of sequence with no indication for what is what. This needs to be clarified.
Answer: Thanks for your advice. We have revised it on according to your advice.
- Ln273: I’m not sure what word the authors are looking for, but I’m pretty sure “handsome” is not it. Please clarify.
Answer: Thanks for your advice. We have revised it on Ln289-290 according to your advice.
- Ln284: I agree that the EM and standard microscope images show that CGVIS can infect and cause cytopathic effect in this cell line, but this study provides no evidence that CGVIS replicates in this cell line. To make that statement, a titre or viral genome quantity would need to be taken at time of infection and then again later to show that titre and/or genome count had increased. Unless this information is provided, please delete or change this sentence for specific accuracy.
Answer: Thanks for your advice. We have revised it on Ln314-315 according to your advice.
- Ln311: the sentence states “significantly higher…” yet I see no statistical support for this statement. Please provide the statistical support. Same with saying the iniae is higher than S. agalactiae.
Answer: Thanks for your advice. We have revised it on Ln196-200 according to your advice.
- Ln349: start a new paragraph with “During the initial passages…”
Answer: Thanks for your advice. We have revised it on Ln367 according to your advice.
- Ln351: would be good to mention what cell type the uniformity has taken on (e.g., figroblast-like?).
Answer: Thanks for your advice. We have added the discussion of the cell morphology on Ln369.
- Ln360: Start a new paragraph with “Generally, the serum is.”
Answer: Thanks for your advice. We have revised it on Ln379 according to your advice.
- Ln368: The discussion regarding different types of media should be expanded. The once sentence here is not enough in my opinion.
Answer: Thanks for your advice. We have added the discussion of different types of media on Ln388-394.
- Ln373: Specify what the number of chromosomes typically is for Tilapia. This can be chased up from the reference but should be stated in the text as well for easy understanding.
Answer: Thanks for your advice. We have revised it on Ln398 according to your advice.
- Ln384: Please expand on the differences typically observed between mammalian cell transfection rates and fish, specifically how your transfection rate in this study compares to other mammals and fish.
Answer: Thanks for your advice. We have revised it in Figure 5 and Ln415-416 according to your advice.
- Ln397: Here would be a good place to explain why CGSIV was specifically chosen to infect this cell line.
Answer: Thanks for your advice. We have revised it on Ln425-429 according to your advice.
- Ln401: Although not necessary as a textual addition, I am curious if TAH-11 has been exposed to TLiV and if the cell line can culture that virus as well. Given its high global economic impact, if the authors have any information, they would like to share this would be a good place to do it.
Answer: Thanks for your advice. We carried out the experiments of TLiV exposure, and observed the obvious CPE phenomenon, but we have not prepared an electron microscopic sample. Next, we will focus on studying the specific mechanism of TLiV infecting TAH-11 cells.
- Ln427: start a new paragraph with “Apoptosis is physiological…”
Answer: Thanks for your advice. We have revised it Ln455 according to your advice.
- Ln433: It is stated that apoptosis related factors (caspases 3 & 9) were upregulated after both agalactiae and S. iniae exposure, but I would argue that biologically speaking S. agalactiae did not have an effect whereas S. iniae did. Caspase 3 was only marginally upregulated (2-fold) which would have questionable biological relevance and Caspase 9 was not altered at al. Contrast this with 6- and 4-fold respective inductions in response to S. iniae which seem slightly more convincing, although still nothing drastic. I think there needs to be some discussion concerning this point, as well as how it relates to other inflammatory heart responses in fish where 20-50-fold inductions are not be uncommon.
Answer: Thanks for your advice. We have revised it Ln462-480 according to your advice.
Reviewer 2 Report
This is an interesting work. The establishment of a new cell line entails an advance in the field, opening new tools for the research. The paper is well written and clearly shows the results as well as the description of this new cell line. However, it contains some typing errors and the description of the materials and methods could be improved. The number of replicates used for each technique is not clear.
Minor comments
Material and methods
-Please, include the ethics discourse and the corresponding code from an ethical committee.
-- please, explain how many replicates were done. You have to include all the data to replicate the experiment.
Line 200: Please, include a part for the statistics analyses. The description of the statistical tools used in this study needs to be improved. For example: if normality was probed, if Anova or T-Student was done, or homogeneity of variances was tested.
Line 203: the efficiency from each gene is described, please introduce.
Line 261: The sequence product should be included in a file, in FASTA format, for allow the access in order to confirm and replicate, following the journals politics and instructions.
Figure 5 line 276: The image without fluorescence should the showed, before the merge. The scale bar should be showed.
Line 308: E. coli was considered as a non pathogenic bacteria, please introduce a cite.
Figure 9 line 337. The legend doesn’t explain completely the figure. Please improve the figure legends, including color, number of replicates…
Author Response
Response to reviewer
Dear Editor and Reviewers:
Thank you very much for your letter and the reviewers' comments on the paper we submitted to the journal of fishes.
The manuscript was carefully revised by using red text according to the comments. We responded point by point to each reviewer’s comments as listed below, along with a clear indication of the location of the revision.
Hope these will make it more acceptable for publication.
Thank you very much again.
Best wishes
Bei Wang
Reviewer #2:
- --Please, include the ethics discourse and the corresponding code from an ethical committee.
Answer: Thanks for your advice. We have added the ethics discourse and the corresponding code from an ethical committee on lines 505 to 507.
- -- please, explain how many replicates were done. You have to include all the data to replicate the experiment.
Answer: Thanks for your advice. We have added the description of the number of experimental replicates on lines 90-91.
- Line 200: Please, include a part for the statistics analyses. The description of the statistical tools used in this study needs to be improved. For example: if normality was probed, if Anova or T-Student was done, or homogeneity of variances was tested.
Answer: Thanks for your advice. We have improved the description of the statistical tools used in this study on lines 212-215.
- Line 203: the efficiency from each gene is described, please introduce.
Answer: Thanks for your advice. We have added the efficiency from each gene to the Table 1.
- Line 261: The sequence product should be included in a file, in FASTA format, for allow the access in order to confirm and replicate, following the journals politics and instructions.
Answer: Thanks for your advice. We have added the sequence product in a file to the Table 1, in FASTA format, for allow the access.
- Figure 5 line 276: The image without fluorescence should the showed, before the merge. The scale bar should be showed.
Answer: Thanks for your advice. We have added the image without fluorescence in Figure 5.
- Line 308: coli was considered as a non-pathogenic bacterium, please introduce a cite.
Answer: Thanks for your advice. We have added a cite to explain E. coli as a non-pathogenic bacterium on line 326.
- Figure 9 line 337. The legend doesn’t explain completely the figure. Please improve the figure legends, including color, number of replicates…
Answer: Thanks for your advice. We have redone the Figure 9 and revised the legend again.